# IMPROVING FEW-SHOT VISUAL CLASSIFICATION WITH UNLABELLED EXAMPLES

## ABSTRACT

We propose a transductive meta-learning method that uses unlabelled instances to improve few-shot image classification performance. Our approach combines a regularized Mahalanobis-distance-based soft k-means clustering procedure with a modified state of the art neural adaptive feature extractor to achieve improved test-time classification accuracy using unlabelled data. We evaluate our method on transductive few-shot learning tasks, in which the goal is to jointly predict labels for query (test) examples given a set of support (training) examples. We achieve new state of the art performance on the Meta-Dataset and the mini-ImageNet and tiered-ImageNet benchmarks.

## 1 INTRODUCTION

Deep learning has revolutionized visual classification, enabled in part by the development of large and diverse sets of curated training data (Szegedy et al., 2014; He et al., 2015; Krizhevsky et al., 2017; Simonyan & Zisserman, 2014; Sornam et al., 2017). However, in many image classification settings, millions of labelled examples are not available; therefore, techniques that can achieve sufficient classification performance with few labels are required. This has motivated research on few-shot learning (Feyjie et al., 2020; Wang & Yao, 2019; Wang et al., 2019; Bellet et al., 2013), which seeks to develop methods for developing classifiers with much smaller datasets. Given a few labelled "support" images per class, a few-shot image classifier is expected to produce labels for a given set of unlabelled "query" images. Typical approaches to few-shot learning adapt a base classifier network to a new support set through various means, such as learning new class embeddings (Snell et al., 2017; Vinyals et al., 2016; Sung et al., 2018), amortized (Requeima et al., 2019; Oreshkin et al., 2018) or iterative (Yosinski et al., 2014) partial adaptation of the feature extractor, and complete fine-tuning of the entire network end-to-end (Ravi & Larochelle, 2017; Finn et al., 2017).

In addition to the standard fully supervised setting, techniques have been developed to exploit additional unlabeled support data (semi-supervision) (Ren et al., 2018) as well as information present in the query set (transduction) (Liu et al., 2018; Kim et al., 2019). In our work, we focus on the transductive paradigm, where the entire query set is labeled at the same time. This allows us to exploit the additional unlabeled data, with the hopes of improving classification performance. Existing transductive few-shot classifiers rely on label propagation from labelled to unlabelled examples in the feature space through either k-means clustering with Euclidean distance (Ren et al., 2018) or message passing in graph convolutional networks (Liu et al., 2018; Kim et al., 2019).

Since few-shot learning requires handling a varying number of classes, an important architectural choice is the final feature to class mapping. Previous methods have used the Euclidean distance (Ren et al., 2018), the absolute difference (Koch et al., 2015), cosine similarity (Vinyals et al., 2016), linear classification (Finn et al., 2017; Requeima et al., 2019) or additional neural network layers (Kim et al., 2019; Sung et al., 2018). Bateni et al. (2020) improved these results by using a class-adaptive Mahalanobis metric. Their method, Simple CNAPS, uses a conditional neural-adaptive feature extractor, along with a regularized Mahalanobis-distance-based classifier. This modification to CNAPS (Requeima et al., 2019) achieves improved performance on the Meta-Dataset benchmark (Triantafillou et al., 2019), only recently surpassed by SUR (Dvornik et al., 2020) and URT (Liu et al., 2020). However, performance suffers in the regime where there are five or fewer support examples available per class.

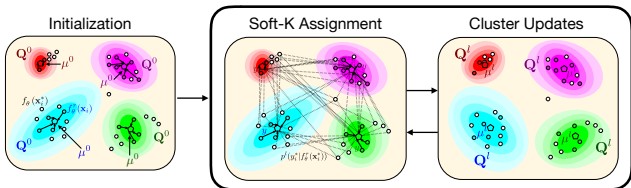

Figure 1: Soft k-means Mahalanobis-distance based clustering method used in Transductive CNAPS. First, cluster parameters are initialized using the support examples. Then, during cluster update iterations, query examples are assigned class probabilities as soft labels and subsequently, both soft-labelled query examples and labelled support examples are used to estimate new cluster parameters.

Motivated by these observations, we explore the use of unlabelled examples through transductive learning within the same framework as Simple CNAPS. Our contributions are as follows. **(1)** We propose a transductive few-shot learner, namely Transductive CNAPS, that extends Simple CNAPS with a transductive two-step task encoder, as well as an iterative soft k-means procedure for refining class parameter estimates (mean and covariance) using both labelled and unlabelled examples. **(2)** We demonstrate the efficacy of our approach by achieving new state of the art performance on Meta-Dataset (Triantafillou et al., 2019). **(3)** When deployed with a feature extractor trained on their respective training sets, Transductive CNAPS achieves state of the art performance on 4 out of 8 settings on mini-ImageNet (Snell et al., 2017) and tiered-Imagenet (Ren et al., 2018), while matching state of the art on another 2. **(4)** When additional non-overlapping classes from ImageNet (Russakovsky et al., 2015) are used to train the feature extractor, Transductive CNAPS is able to leverage this example-rich feature extractor to achieve state of the art across the board on mini-ImageNet and tiered-ImageNet.

## 2 RELATED WORK

### 2.1 FEW-SHOT LEARNING USING LABELLED DATA

Early work on few-shot visual classification has focused on improving classification accuracy through the use of better classification metrics with a meta-learned non-adaptive feature extractor. Matching networks (Vinyals et al., 2016) use cosine similarities over feature vectors produced by independently learned feature extractors. Siamese networks (Koch et al., 2015) classify query images based on the nearest support example in feature space, under the $L_1$ metric. Relation networks (Sung et al., 2018) and variants (Kim et al., 2019; Satorras & Estrach, 2018) learn their own similarity metric, parameterised through a Multi-Layer Perceptron. More recently, Prototypical Networks (Snell et al., 2017) learn a shared feature extractor that is used to produce class means in a feature space where the Euclidean distance is used for classification.

Other work has focused on adapting the feature extractor for new tasks. Transfer learning by fine-tuning pretrained visual classifiers (Yosinski et al., 2014) was an early approach that proved limited in success due to issues arising from over-fitting. MAML (Finn et al., 2017) and its variants (Mishra et al., 2017; Nichol et al., 2018; Ravi & Larochelle, 2017) learn meta-parameters that allow fast task-adaptation with only a few gradient updates. Work has also been done on partial adaptation of feature extractors using conditional neural adaptive processes (Oreshkin et al., 2018; Garnelo et al., 2018; Requeima et al., 2019; Bateni et al., 2020). These methods rely on channel-wise adaptation of pretrained convolutional layers by adjusting parameters of FiLM layers (Perez et al., 2018) inserted throughout the network. Our work builds on the most recent of these neural adaptive approaches, specifically Simple CNAPS (Bateni et al., 2020). SUR (Dvornik et al., 2020) and URT (Liu et al., 2020) are two very recent methods that employ universal representations stemming from multiple domain-specific feature extraction heads. URT (Liu et al., 2020), which was developed and released publicly in parallel to this work, achieves state of the art performance by using a universal transformation layer.

### 2.2 FEW-SHOT LEARNING USING UNLABELLED DATA

Several approaches (Kim et al., 2019; Liu et al., 2018; Ren et al., 2018) have also explored the use of unlabelled instances for few-shot visual classification. EGNN (Kim et al., 2019) employs a

Figure 2: Overview of the neural adaptive feature extraction process used in Transductive/Simple CNAPS. Figure was adapted from Bateni et al. (2020).

graph convolutional edge-labelling network for iterative propagation of labels from support to query instances. Similarly, TPN (Liu et al., 2018) learns a graph construction module for neural propagation of soft labels between elements of the query set. These methods rely on a neural parameterization of distance within the feature space. TEAM (Qiao et al., 2019) uses an episodic-wise transductive adaptable metric for performing inference on query examples using a task-specific metric. Song et al. (2020) use a cross attention network combined with a transductive iterative approach for augmenting the support set using the query examples. The closest method to our work is Ren et al. (2018). Their approach extends prototypical networks by performing a single additional soft-label weighted estimation of class prototypes. Our work, on the other hand, is different in three major ways. First, we produce soft-labelled estimates of both class mean and covariance. Second, we use an iterative algorithm with a data-driven convergence criterion allowing for a dynamic number of soft-label updates, depending on the task at hand. Lastly, we employ a neural adaptive procedure for feature extraction that is conditioned on a two-step learned transductive task representation, as opposed to a fixed feature-extractor. As we discuss in Section 4.2, this novel task-representation encoder is responsible for substantial performance gains on out-of-domain tasks.

## 3 METHOD

### 3.1 PROBLEM DEFINITION

Following (Snell et al., 2017; Bateni et al., 2020; Requeima et al., 2019; Finn et al., 2017), we focus on a few-shot classification setting where a distribution $D$ over image classification tasks $(\mathcal{S}, \mathcal{Q})$ is provided for training. Each task $(\mathcal{S}, \mathcal{Q}) \sim D$ consists of a support set $\mathcal{S} = \{(\mathbf{x}_i, y_i)\}_{i=1}^n$ of labelled images and a query set $\mathcal{Q} = \{\mathbf{x}_i^*\}_{i=1}^m$ of unlabelled images; the goal is to predict labels for these query examples, given the (typically small) support set. Each query image $\mathbf{x}_i^* \in \mathcal{Q}$ has a corresponding ground truth label $y_i^*$ available at training time. A model will be trained by minimizing, over some parameters $\theta$ (which are shared across tasks), the expected query set classification loss over tasks: $\mathbb{E}_{(\mathcal{S}, \mathcal{Q}) \sim D}[\sum_{\mathbf{x}_i^* \in \mathcal{Q}} - \log p_\theta(y_i^* | \mathbf{x}_i^*, \mathcal{S}, \mathcal{Q})]$; the inclusion of the dependence on all of $\mathcal{Q}$ here allows for the model to be transductive. At test time, a separate distribution of tasks generated from previously unseen images and classes is used to evaluate performance. We also define *shot* as the number of support examples per class, and *way* as the number of classes within the task.

### 3.2 SIMPLE CNAPS

Our method extends the Simple CNAPS (Bateni et al., 2020) architecture for few-shot visual classification. Simple CNAPS performs few-shot classification in two steps. First, it computes task-adapted features for every support and query example. This part of the architecture is the same as that in CNAPS (Requeima et al., 2019), and is based on the FiLM meta-learning framework (Perez et al., 2018). Second, it uses the support set to estimate a per-class Mahalanobis metric, which is used to assign query examples to classes. The architecture uses a ResNet18 (He et al., 2015) feature extractor. Within each residual block, Feature-wise Linear Modulation (FiLM) layers compute a scale factor $\gamma$ and shift $\beta$ for each output channel, using block-specific adaptation networks $\psi_\theta$ that are conditioned on a task encoding. The task encoding $g_\theta(\mathcal{S})$ consists of the mean-pooled feature vectors of support examples produced by $d_\theta$, a separate but end-to-end learned Convolution Neural Network (CNN). This produces an adapted feature extractor $f_\theta$ (which implicitly depends on the support set $\mathcal{S}$) that maps support/query images onto the corresponding adapted feature space. We will denote by $\mathcal{S}_\theta, \mathcal{Q}_\theta$ versions of the support/query sets where each image is mapped into its feature representation $\mathbf{z} = f_\theta(\mathbf{x})$. Simple CNAPS then computes a Mahalanobis distance relative to each

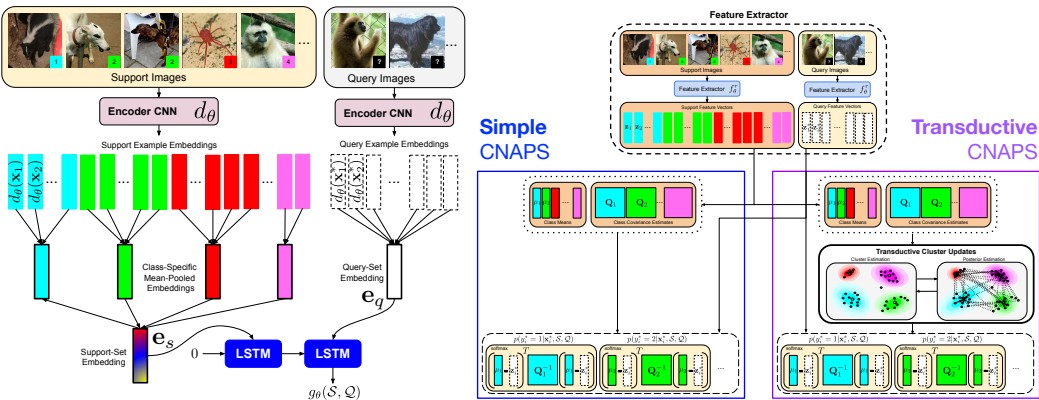

(a) Transductive Set-Encoder Overview    (b) Transductive CNAPS vs. Simple CNAPS

Figure 3: a) Overview of the transductive task-encoding procedure, $g_\theta(\mathcal{S}, \mathcal{Q})$, used in Transductive CNAPS. b) Transductive CNAPS (right) extends the Mahalanobis-distance based classifier in Simple CNAPS (left) through transductive soft k-means clustering of the visual space.

class $k$ by estimating a mean $\boldsymbol{\mu}_k$ and regularized covariance $\mathbf{Q}_k$ in the adapted feature space, using the support instances:

$$\boldsymbol{\mu}_k = \frac{1}{n_k} \sum_i \mathbb{I}[y_i = k]\, \mathbf{z}_i, \qquad \mathbf{Q}_k = \lambda_k\, \boldsymbol{\Sigma}_k + (1 - \lambda_k)\, \boldsymbol{\Sigma} + \beta I, \qquad \lambda_k = \frac{n_k}{n_k + 1}. \quad (1)$$

Here $\mathbb{I}[y_i = k]$ is the indicator function and $n_k = \sum_i \mathbb{I}[y_i = k]$ is the number of examples with class $k$ in the support set $\mathcal{S}$. The ratio $\lambda_k$ balances a task-conditional sample covariance $\boldsymbol{\Sigma}$ and a class-conditional sample covariance $\boldsymbol{\Sigma}_k$:

$$\boldsymbol{\Sigma} = \frac{1}{n} \sum_i \big(\mathbf{z}_i - \boldsymbol{\mu}\big)\big(\mathbf{z}_i - \boldsymbol{\mu}\big)^T, \qquad \boldsymbol{\Sigma}_k = \frac{1}{n_k} \sum_i \mathbb{I}[y_i = k] \big(\mathbf{z}_i - \boldsymbol{\mu}_k\big)\big(\mathbf{z}_i - \boldsymbol{\mu}_k\big)^T, \quad (2)$$

where $\boldsymbol{\mu} = \frac{1}{n} \sum_i \mathbf{z}_i$ is the task-level mean. When few support examples are available for a particular class, $\lambda_k$ is small, and the estimate is regularized towards the task-level covariance $\boldsymbol{\Sigma}$. As the number of support examples for the class increases, the estimate tends towards the class-conditional covariance $\boldsymbol{\Sigma}_k$. Additionally, a regularizer $\beta I$ (we set $\beta = 1$ in our experiments) is added to ensure invertibility. Given the class means and covariances, Simple CNAPS computes class probabilities for each query feature vector $\mathbf{z}_i^*$ through a softmax over the squared Mahalanobis distances with respect to each class:

$$p(y^* = k \mid \mathbf{z}^*) \propto \exp\big(-(\mathbf{z} - \boldsymbol{\mu}_k)^T \mathbf{Q}_k^{-1}(\mathbf{z} - \boldsymbol{\mu}_k)\big). \quad (3)$$

### 3.3 TRANSDUCTIVE CNAPS

Transductive CNAPS extends Simple CNAPS by taking advantage of the query set, both in the feature adaptation step and the classification step. First, the task encoder $g_\theta$ is extended to incorporate both a support-set embedding $\mathbf{e}_s$ and a query-set embedding $\mathbf{e}_q$ such that,

$$\mathbf{e}_s = \frac{1}{K} \sum_k \frac{1}{n_k} \sum_i \mathbb{I}[y_i = k]\, d_\theta(\mathbf{x}_i), \qquad \mathbf{e}_q = \frac{1}{n_q} \sum_{i*} d_\theta(\mathbf{x}_i^*), \quad (4)$$

where $d_\theta$ is a learned CNN. The support embedding $\mathbf{e}_s$ is formed by an average of (encoded) support examples, with weighting inversely proportional to their class counts to prevent bias from class imbalance. The query embedding $e_q$ uses simple mean-pooling; both $\mathbf{e}_s$ and $\mathbf{e}_q$ are invariant to permutations of the respective support/query instances. We then process $e_s$ and $e_q$ through two steps of a Long Short Term Memory (LSTM) network in the same order to generate the final transductive task-embedding $g_\theta(\mathcal{S}, \mathcal{Q})$ used for adaptation. This process is visualized in Figure 3-a.

---

**Algorithm 1** Iterative Refinement in Transductive-CNAPS

---

1: **procedure** COMPUTE_QUERY_LABELS($\mathcal{S}_\theta, \mathcal{Q}_\theta, N_{\text{iter}}$)
2:     For $j$ ranging over support and query sets, $w_{jk} \leftarrow \begin{cases} 1 & \text{if } (\mathbf{z}'_j, y'_j) \in \mathcal{S}_\theta \text{ and } y_j = k \\ 0 & \text{otherwise} \end{cases}$
3:     **for** iter = $0 \cdots N_{\text{iter}}$ **do**             ▷ The first iteration is equivalent to Simple CNAPS;
4:         Compute class parameters $\boldsymbol{\mu}_k, \mathbf{Q}_k$ according to update equations equation 6-equation 7
5:         Compute class weights using class parameters according to equation 5
6:         **break** if the most probable class for each query example hasn't changed
7:     **end for**
8:     **return** class probabilities $w_{jk}$ for $j$ corresponding to $\mathcal{Q}_\theta$
9: **end procedure**

---

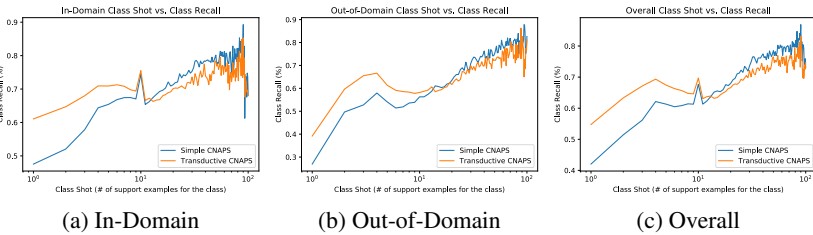

(a) In-Domain             (b) Out-of-Domain            (c) Overall

Figure 4: Class recall (otherwise noted as in-class query accuracy) averaged between classes across all tasks and (a: In-Domain, b: Out-of-domain, c: all) Meta-Dataset datasets. Class recalls have been grouped together, averaged and plotted according to the class shot in (a), (b), and (c).

Second, we can interpret Simple CNAPS as a form of supervised clustering in feature space; each cluster (corresponding to a class $k$) is parameterized with a centroid $\boldsymbol{\mu}_k$ and a metric $\mathbf{Q}_k^{-1}$, and we interpret equation 3 as class assignment probabilities based on the distance to each centroid. With this viewpoint in mind, a natural extension to consider is to use the estimates of the class assignment probabilities on unlabelled data to refine the class parameters $\boldsymbol{\mu}_k, \mathbf{Q}_k$ in a soft $k$-means framework based on per-cluster Mahalanobis distances (Melnykov & Melnykov, 2014). In this framework, as shown in Figure 1, we alternate between computing updated assignment probabilities using equation 3 on the query set and using those assignment probabilities to compute updated class parameters.

We will define $\mathcal{R}_\theta = \mathcal{S}_\theta \sqcup \mathcal{Q}_\theta$ as the disjoint union of the support set and the query set. For each element of $\mathcal{R}_\theta$, which we index by $j$, we define responsibilities $w_{jk}$ in terms of their class predictions when it is part of the query set and in terms of the label when it is part of the support set,

$$w_{jk} = \begin{cases} p(y'_j = k \mid \mathbf{z}'_j) & \mathbf{z}'_j \in \mathcal{Q}_\theta, \\ \mathbb{I}[y'_j = k] & (\mathbf{z}'_j, y'_j) \in \mathcal{S}_\theta. \end{cases} \tag{5}$$

Using these responsibilities we can incorporate unlabelled samples from the support set by defining weighted estimates $\boldsymbol{\mu}'_k$ and $\mathbf{Q}'_k$:

$$\boldsymbol{\mu}'_k = \frac{1}{n'_k} \sum_j w_{jk} \, \mathbf{z}'_j \qquad\qquad \mathbf{Q}'_k = \lambda'_k \boldsymbol{\Sigma}'_k + (1 - \lambda'_k)\boldsymbol{\Sigma}' + \beta I, \tag{6}$$

where $n'_k = \sum_j w_{jk}$ defines $\lambda'_k = n'_k/(n'_k + 1)$, and the covariance estimates $\boldsymbol{\Sigma}'$ and $\boldsymbol{\Sigma}'_k$ are

$$\boldsymbol{\Sigma}' = \frac{1}{\sum_k n'_k} \sum_{jk} w_{jk} (\mathbf{z}'_j - \boldsymbol{\mu}')(\mathbf{z}'_j - \boldsymbol{\mu}')^T, \qquad \boldsymbol{\Sigma}'_k = \frac{1}{n'_k} \sum_j w_{jk} (\mathbf{z}'_j - \boldsymbol{\mu}'_k)(\mathbf{z}'_j - \boldsymbol{\mu}'_k)^T. \tag{7}$$

with $\boldsymbol{\mu}' = (\sum_k n'_k)^{-1} \sum_{jk} w_{jk}\mathbf{z}'_j$ being the task-level mean.

These update equations are simply weighted versions of the original Simple CNAPS estimators from Section 3.2, and reduce to them exactly in the case of an empty query set.

Algorithm 1 summarizes the soft k-means procedure based on these updates. We initialize our weights using only the labelled support set. We use those weights to compute class parameters, then compute

updated weights using both the support and query sets. At this point, the weights associated with the query set $\mathcal{Q}$ are the same class probabilities as estimated by Simple CNAPS. However, we continue this procedure iteratively until we reach either reach a maximum number of iterations, or until class assignments $\mathrm{argmax}_k\, w_{jk}$ stop changing.

Unlike the transductive task-encoder, this second extension, namely the soft k-mean iterative estimation of class parameters, is used at test time only. During training, a single estimation is produced for both mean and covariance using only the support examples. This, as we discuss more in Section 4.2, was shown to empirically perform better. See Figure 3-b for a high-level visual comparison of classification in Simple CNAPS vs. Transductive CNAPS.

## 4 EXPERIMENTS

### 4.1 BENCHMARKS

**Meta-Dataset** (Triantafillou et al., 2019) is a few-shot image classification benchmark that consists of 10 widely used datasets: ILSVRC-2012 (ImageNet) (Russakovsky et al., 2015), Omniglot (Lake et al., 2015), FGVC-Aircraft (Aircraft) (Maji et al., 2013), CUB-200-2011 (Birds) (Wah et al., 2011), Describable Textures (DTD) (Cimpoi et al., 2014), QuickDraw (Jongejan et al., 2016), FGVCx Fungi (Fungi) (Schroeder & Cui, 2018), VGG Flower (Flower) (Nilsback & Zisserman, 2008), Traffic Signs (Signs) (Houben et al., 2013) and MSCOCO (Lin et al., 2014). Consistent with past work (Requeima et al., 2019; Bateni et al., 2020), we train our model on the official training splits of the first 8 datasets and use the test splits to evaluate in-domain performance. We use the remaining two dataset as well as three external benchmarks, namely MNIST (LeCun & Cortes, 2010), CIFAR10 (Krizhevsky, 2009) and CIFAR100 (Krizhevsky, 2009), for out-of-domain evaluation. Task generation in Meta-Dataset follows a complex procedure where tasks can be of different *ways* and individual classes can be of varying *shots* even within the same task. Specifically, for each task, the task *way* is first sampled uniformly between 5 and 50 and *way* classes are selected at random from the corresponding class/dataset split. Then, for each class, 10 instances are sampled at random and used as query examples for the class, while of the remaining images for the class, a *shot* is sampled uniformly from [1, 100] and *shot* number of images are selected at random as support examples with total support set size of 500. Additional dataset-specific constraints are enforced, as discussed in Section 3.2 of (Triantafillou et al., 2019), and since some datasets have fewer than 50 classes and fewer than 100 images per class, the overall *way* and *shot* distributions resemble Poisson distributions where most tasks have fewer than 10 classes and most classes have fewer than 10 support examples (see Appendix-A.1). Following Bateni et al. (2020) and Requeima et al. (2019), we first train our ResNet18 feature extractor on the Meta-Dataset defined training split of ImageNet following the procedure in Appendix-A.3. The ResNet18 parameters are then kept fixed while we train the adaptation network for a total of sampled 110K tasks using Episodic Training (Snell et al., 2017; Finn et al., 2017) (see Appendix-A.3 for details).

**mini/tiered-ImageNet** (Vinyals et al., 2016; Ren et al., 2018) are two benchmarks for few-shot learning. Both datasets employ subsets of ImageNet (Russakovsky et al., 2015) with a total of 100 classes and 60K images in mini-ImageNet and 608 classes and 779K images in tiered-ImageNet. Unlike Meta-Dataset, tasks across these datasets have pre-defined *shots*/*ways* that are uniform across every task generated in the specified setting. Following (Nichol et al., 2018; Liu et al., 2018; Snell et al., 2017), we report performance on the 1/5-*shot* 5/10-*way* settings across both datasets with 10 query examples per class. We first train the ResNet18 on the training set of the corresponding benchmark at hand following the procedure noted in Appendix-A.4. We also consider a more feature-rich ResNet18 trained on the larger ImageNet dataset. However, we exclude classes and examples from test sets of mini/tiered-ImageNet to address potential class/example overlap issues, resulting in 825 classes and 1,055,494 images remaining. Then, with the ResNet18 parameters fixed, we train episodically for 20K tasks (see Appendix-A.2 for details).

### 4.2 RESULTS

**Evaluation on Meta-Dataset:** In-domain, out-of-domain and overall rankings on Meta-Dataset are shown in Table 1. Following Bateni et al. (2020) and Requeima et al. (2019), we pretrain our ResNet feature extractor on the training split of the ImageNet subset of Meta-Dataset. As demonstrated,

| | In-Domain Accuracy (%) | | | | | | | | Out-of-Domain Accuracy (%) | | | | | Avg Rank | | |
|---|---|---|---|---|---|---|---|---|---|---|---|---|---|---|---|---|
| Model | ImageNet | Omniglot | Aircraft | Birds | DTD | QuickDraw | Fungi | Flower | Signs | MSCOCO | MNIST | CIFAR10 | CIFAR100 | In | Out | All |
| RelationNet | 30.9±0.9 | 86.6±0.8 | 69.7±0.8 | 54.1±1.0 | 56.6±0.7 | 61.8±1.0 | 32.6±1.1 | 76.1±0.8 | 37.5±0.9 | 27.4±0.9 | NA | NA | NA | 10.5 | 11.0 | 10.6 |
| MatchingNet | 36.1±1.0 | 78.3±1.0 | 69.2±1.0 | 56.4±1.0 | 61.8±0.7 | 60.8±1.0 | 33.7±1.0 | 81.9±0.7 | 55.6±1.1 | 28.8±1.0 | NA | NA | NA | 10.1 | 8.5 | 9.8 |
| MAML | 37.8±1.0 | 83.9±1.0 | 76.4±0.7 | 62.4±1.1 | 64.1±0.8 | 59.7±1.1 | 33.5±1.1 | 79.9±0.8 | 42.9±1.3 | 29.4±1.1 | NA | NA | NA | 9.2 | 10.5 | 9.5 |
| ProtoNet | 44.5±1.1 | 79.6±1.1 | 71.1±0.9 | 67.0±1.0 | 65.2±0.8 | 64.9±0.9 | 40.3±1.1 | 86.9±0.7 | 46.5±1.0 | 39.9±1.1 | NA | NA | NA | 8.2 | 9.5 | 8.5 |
| ProtoMAML | 46.5±1.1 | 82.7±1.0 | 75.2±0.8 | 69.9±1.0 | 68.3±0.8 | 66.8±0.9 | 42.0±1.2 | 88.7±0.7 | 52.4±1.1 | 41.7±1.1 | NA | NA | NA | 7.1 | 8.0 | 7.3 |
| CNAPS | 52.3±1.0 | 88.4±0.7 | 80.5±0.6 | 72.2±0.9 | 58.3±0.7 | 72.5±0.8 | 47.4±1.0 | 86.0±0.5 | 60.2±0.9 | 42.6±1.1 | 92.7±0.4 | 61.5±0.7 | 50.1±1.0 | 6.6 | 6.0 | 6.4 |
| BOHB-E | 55.4±1.1 | 77.5±1.1 | 60.9±0.9 | 73.6±0.8 | 72.8±0.7 | 61.2±0.9 | 44.5±1.1 | 90.6±0.6 | 57.5±1.0 | 51.9±1.0 | NA | NA | NA | 6.4 | 4.0 | 5.9 |
| TaskNorm | 50.6±1.1 | 90.7±0.6 | 83.8±0.6 | 74.6±0.8 | 62.1±0.7 | 74.8±0.7 | 48.7±1.0 | 89.6±0.5 | 67.0±0.7 | 43.4±1.0 | 92.3±0.4 | 69.3±0.8 | 54.6±1.1 | 4.7 | 4.8 | 4.8 |
| Simple CNAPS | **58.6±1.1** | 91.7±0.6 | 82.4±0.7 | 74.9±0.8 | 67.8±0.8 | 77.7±0.7 | 46.9±1.0 | 90.7±0.5 | 73.5±0.7 | 46.2±1.1 | 93.9±0.4 | 74.3±0.7 | 60.5±1.0 | 3.4 | 3.0 | 3.2 |
| SUR | 56.3±1.1 | 93.1±0.5 | **85.4±0.7** | 71.4±1.0 | **71.5±0.8** | 81.3±0.6 | **63.1±1.0** | 82.8±0.7 | 70.4±0.8 | **52.4±1.1** | 94.3±0.4 | 66.8±0.9 | 56.6±1.0 | 3.1 | 2.6 | 2.9 |
| URT | 55.7±1.0 | **94.4±0.4** | **85.8±0.6** | 76.3±0.8 | **71.8±0.7** | **82.5±0.6** | **63.5±1.0** | 88.2±0.6 | 69.4±0.8 | **52.2±1.1** | 94.8±0.4 | 67.3±0.8 | 56.9±1.0 | **1.7** | 2.8 | 2.2 |
| Our Method | **58.8±1.1** | **93.9±0.4** | 84.1±0.6 | **76.8±0.8** | 69.0±0.8 | 78.6±0.7 | 48.8±1.1 | **91.6±0.4** | **76.1±0.7** | 48.7±1.0 | **95.7±0.3** | **75.7±0.7** | **62.9±1.0** | 2.1 | **1.6** | **1.9** |

Table 1: Few-shot classification on Meta-Dataset, MNIST, and CIFAR10/100. Error intervals showcase 95% confidence interval, and bold values indicate statistically significant state of the art performance. Average rank is obtained by ranking methods on each dataset and averaging the ranks.

| | | mini-ImageNet Accuracy (%) | | | | tiered-ImageNet Accuracy (%) | | | |
|---|---|---|---|---|---|---|---|---|---|
| | | 5-*way* | | 10-*way* | | 5-*way* | | 10-*way* | |
| Model | Transductive | 1-*shot* | 5-*shot* | 1-*shot* | 5-*shot* | 1-*shot* | 5-*shot* | 1-*shot* | 5-*shot* |
| MAML (Finn et al., 2017) | BN | 48.7±1.8 | 63.1±0.9 | 31.3±1.1 | 46.9±1.2 | 51.7±1.8 | 70.3±1.7 | 34.4±1.2 | 53.3±1.3 |
| MAML+ (Liu et al., 2018) | Yes | 50.8±1.8 | 66.2±1.8 | 31.8±0.4 | 48.2±1.3 | 53.2±1.8 | 70.8±1.8 | 34.8±1.2 | 54.7±1.3 |
| Reptile (Nichol et al., 2018) | No | 47.1±0.3 | 62.7±0.4 | 31.1±0.3 | 44.7±0.3 | 49.0±0.2 | 66.5±0.2 | 33.7±0.3 | 48.0±0.3 |
| Reptile+BN (Nichol et al., 2018) | BN | 49.9±0.3 | 66.0±0.6 | 32.0±0.3 | 47.6±0.3 | 52.4±0.2 | 71.0±0.2 | 35.3±0.3 | 52.0±0.3 |
| ProtoNet (Snell et al., 2017) | No | 46.1±0.8 | 65.8±0.7 | 32.9±0.5 | 49.3±0.4 | 48.6±0.9 | 69.6±0.7 | 37.3±0.6 | 57.8±0.5 |
| RelationNet (Sung et al., 2018) | BN | 51.4±0.8 | 67.0±0.7 | 34.9±0.5 | 47.9±0.4 | 54.5±0.9 | 71.3±0.8 | 36.3±0.6 | 58.0±0.6 |
| TPN (Liu et al., 2018) | Yes | 51.4±0.8 | 67.1±0.7 | 34.9±0.5 | 47.9±0.4 | 59.9±0.9 | 73.3±0.7 | 44.8±0.6 | 59.4±0.5 |
| AttWeightGen (Gidaris & Komodakis, 2018) | BN | 56.2±0.9 | 73.0±0.6 | NA | NA | NA | NA | NA | NA |
| TADAM (Oreshkin et al., 2018) | BN | 58.5±0.3 | 76.7±0.3 | NA | NA | NA | NA | NA | NA |
| Simple CNAPS (Bateni et al., 2020) | BN | 53.2±0.9 | 70.8±0.7 | 37.1±0.5 | 56.7±0.5 | 63.0±1.0 | 80.0±0.8 | 48.1±0.7 | 70.2±0.6 |
| LEO (Rusu et al., 2018) | BN | 61.8±0.1 | 77.6±0.1 | NA | NA | 66.3±0.1 | 81.4±0.1 | NA | NA |
| Transductive CNAPS | Yes | 55.6±0.9 | 73.1±0.7 | 42.8±0.7 | 59.6±0.5 | 65.9±1.0 | 81.8±0.7 | 54.6±0.8 | 72.5±0.6 |
| Simple CNAPS (Bateni et al., 2020) + FETI | BN | 77.4±0.8 | 90.3±0.4 | 63.5±0.6 | 83.1±0.4 | 71.4±1.0 | 86.0±0.6 | 57.1±0.7 | 78.5±0.5 |
| Transductive CNAPS + FETI | Yes | **79.9±0.8** | **91.5±0.4** | **68.5±0.6** | **85.9±0.3** | **73.8±1.0** | **87.7±0.6** | **65.1±0.8** | **80.6±0.5** |

Table 2: Few-shot visual classification results on 1/5-shot 5/10-way few-shot on mini/tiered-ImageNet. For CNAP-based model, "FETI" indicates that the feature extractor used has been trained on ImageNet Russakovsky et al. (2015) exluding classes within the test splits of mini/tiered-ImageNet (for more details see Appendix-[TBD]). "BN" indicates implicit transductive conditioning on the query set through the use of batch normalization. Error intervals showcase 95% confidence interval.

Transductive CNAPS sets new state of the art accuracy on 2 out of the 8 in-domain datasets, while matching other methods on 2 of the remaining domains. On out-of-domain tasks, it performs better with new state of the art performance on 4 out of the 5 out-of-domain datasets, Overall, it produces an average rank of 1.9 among all datasets, the best among the methods, with an average rank of 2.1 on in-domain tasks, only second to URT which was developed parallel to Transductive CNAPS, and 1.6 on out-of-domain tasks, the best among even the most recent methods.

**Evaluation on mini/tiered-ImageNet:** We consider two feature extractor training settings on these benchmarks. First, we employ the feature extractor trained on the corresponding training split of the mini/tiered-ImageNet. As shown in Table 2, on tiered-ImageNet, Transductive CNAPS achieves state of art performance on both 10-way settings while matching state of the art accuracy of LEO (Rusu et al., 2018) on the 5-way settings. On mini-ImageNet, Transductive CNAPS out-performs other methods on 10-way settings while coming second to LEO (Rusu et al., 2018) and TADAM (Requeima et al., 2019; Oreshkin et al., 2018) on 5-way. We attribute this difference in performance between mini-ImageNet and tiered-ImageNet to the fact that mini-ImageNet only provides 38,400 training examples, compared to 448,695 examples provided by tiered-ImageNet. This results in a lower performing ResNet-18 feature extractor (which is trained in a traditional supervised manner). This hypothesis is further supported by the results provided in our second model (denoted by "FETI", for "Feature Extractor Trained with ImageNet", in Table 2). In this model, we train the feature extractor with a much larger subset of ImageNet, which has been carefully selected to prevent any possible overlap (in examples or classes) with the test sets of mini/tiered-ImageNet. In this case, Transductive CNAPS is able to take advantage of the more example-rich feature extractor, establishing state of the art performance across the board. Additionally, even when compared to Simple CNAPS using the

| CNAPS Model | In-Domain Accuracy (%) | | | | | | | | Out-of-Domain Accuracy (%) | | | | | Avg Acc. | | |
|---|---|---|---|---|---|---|---|---|---|---|---|---|---|---|---|---|
| | ImageNet | Omniglot | Aircraft | Birds | DTD | QuickDraw | Fungi | Flower | Signs | MSCOCO | MNIST | CIFAR10 | CIFAR100 | In | Out | All |
| GMM-EM+ | 53.3±1.0 | 91.8±0.6 | 81.2±0.6 | **75.8±0.7** | **71.8±0.6** | 72.9±0.7 | 42.8±0.9 | 91.0±0.4 | 66.1±0.8 | 40.3±1.0 | 94.2±0.4 | 69.0±0.7 | 51.3±0.9 | 72.6 | 64.2 | 69.3 |
| GMM | 45.3±1.0 | 88.0±0.9 | 80.8±0.8 | 71.4±0.8 | 61.1±0.7 | 70.7±0.8 | 42.9±1.0 | 88.1±0.6 | 68.9±0.7 | 37.2±0.9 | 91.4±0.5 | 64.5±0.7 | 46.6±0.9 | 68.5 | 61.7 | 65.9 |
| FEOT GMM | 52.6±1.1 | 89.6±0.7 | **84.0±0.6** | 76.2±0.6 | 66.5±0.8 | 73.4±0.8 | 45.7±1.0 | 89.8±0.6 | 74.4±0.7 | 44.2±1.0 | 93.1±0.4 | 71.1±0.8 | 56.9±1.0 | 72.2 | 67.9 | 70.6 |
| COT GMM | 48.7±1.0 | 92.3±0.5 | 80.0±0.7 | 72.4±0.7 | 59.8±0.7 | 71.1±0.7 | 41.4±0.9 | 87.7±0.5 | 63.6±0.8 | 39.2±0.8 | 89.8±0.5 | 66.9±0.7 | 50.5±0.8 | 69.2 | 62.0 | 66.4 |
| GMM-EM | 52.3±1.0 | 92.0±0.5 | **84.3±0.6** | 75.2±0.8 | 64.3±0.7 | 72.6±0.8 | 44.6±1.0 | 90.8±0.5 | 71.4±0.7 | 44.7±0.9 | 93.0±0.4 | 71.1±0.7 | 56.4±0.9 | 72.0 | 67.3 | 70.2 |
| Transductive+ | 53.3±1.1 | 92.3±0.5 | 81.2±0.7 | 75.0±0.8 | **72.0±0.7** | 74.8±0.8 | 45.1±1.0 | 92.1±0.4 | 71.0±0.8 | 44.0±1.1 | 95.9±0.3 | 71.1±0.7 | 57.3±1.1 | 73.2 | 67.9 | 71.2 |
| Simple | **58.6±1.1** | 91.7±0.6 | 82.4±0.7 | 74.9±0.8 | 67.8±0.8 | 77.7±0.7 | 46.9±1.0 | 90.7±0.5 | 73.5±0.7 | 46.2±1.1 | 93.9±0.4 | 74.3±0.7 | 60.5±1.0 | 73.8 | 69.7 | 72.2 |
| FEOT | **57.3±1.1** | 90.5±0.7 | 82.9±0.7 | 74.8±0.8 | 67.3±0.8 | 76.3±0.8 | 47.7±1.0 | 90.5±0.5 | **75.8±0.7** | **47.1±1.1** | 94.9±0.4 | 74.3±0.8 | **61.2±1.0** | 73.4 | 70.7 | 72.4 |
| COT | **58.8±1.1** | **95.2±0.3** | **84.0±0.6** | 76.4±0.7 | 68.5±0.8 | **77.8±0.7** | 49.7±1.0 | **92.7±0.4** | 70.8±0.7 | **47.3±1.0** | 94.2±0.4 | **75.2±0.7** | **61.2±1.0** | **75.4** | 69.7 | 73.2 |
| Transductive | **58.8±1.1** | 93.9±0.4 | **84.1±0.6** | 76.8±0.8 | 69.0±0.8 | **78.6±0.7** | 48.8±1.1 | 91.6±0.4 | **76.1±0.7** | **48.7±1.0** | 95.7±0.3 | 75.7±0.7 | **62.9±1.0** | 75.2 | **71.8** | **73.9** |

Table 3: Performance of various ablations of Tranductive and Simple CNAPS on Meta-Dataset. Error intervals showcase 95% confidence interval, and bold values indicate statistically significant state of the art performance.

same example-rich feature extractor, it outperforms the baseline with strong margins; this comparison demonstrates the gains we get from leveraging the additional query set information.

**Performance vs. Class Shot:** In Figure 4, we examine the relationship between class recall (i.e. accuracy among query examples belonging to the class itself) and the number of support examples in the class (shot). As shown, Transductive CNAPS is very effective when class shot is below 10, showing large average recall improvements, especially at the 1-shot level. However, as the class shot increases beyond 10, performance drops compared to Simple CNAPS. This suggests that soft k-means learning of cluster parameters can be effective when very few support examples are available. Conversely, in high-shot classes, transductive updates can act as distractors.

**Training with Classification-Time Soft K-means Clustering:** In our work, we use soft k-means iterative updates of means and covariance at test-time only. It's natural to consider training the feature adaptation network end-to-end through the soft k-means transduction procedure. We provide this comparison in the bottom-half of Table 3, with "Transductive+ CNAPS" denoting this variation. Iterative updates during training result in an average accuracy decrease of 2.5%, which we conjecture to be due to training instabilities caused by applying this iterative algorithm early in training on noisy features.

**Transductive Feature Extraction vs. Classification:** Our approach extends Simple CNAPS in two ways: improved adaptation of the feature extractor using a transductive task-encoding, and the soft k-means iterative estimation of class means and covariances. We perform two ablations, "Feature Extraction Only Transductive" (FEOT) and "Classification Only Transductive" (COT), to independently assess the impact of these extensions. The results are presented in Table 3. As shown, both extensions outperform Simple CNAPS. The transductive task-encoding is especially effective on out-of-domain tasks whereas the soft k-mean learning of class parameters boosts accuracy on in-domain tasks. Transductive CNAPS is able to leverage the best of both worlds, allowing it to achieve statistically significant gains over Simple CNAPS overall.

**Comparison to Gaussian Mixture Models:** The Mahalanobis-distance based class probabilities produced by Equation 3 closely resembles the cluster posterior probabilities (responsibilities) inferred by a Gaussian Mixture Model (GMM). The only changes required to make this correspondence exact is to introduce a class prior distribution $\pi$, and to change the class probability model equation 3 to the Gaussian likelihood:

$$p(y^* = k \mid \mathbf{z}^*) \propto \pi(y^* = k) \exp\left( -\frac{1}{2}(\mathbf{z} - \boldsymbol{\mu_k})^T \mathbf{Q}_k^{-1}(\mathbf{z} - \boldsymbol{\mu_k}) - \frac{1}{2}\log|\mathbf{Q}_k| \right) \tag{8}$$

With these modifications, Transductive CNAPS would exactly correspond to inference in a GMM, with cluster parameters learned through semi-supervised expectation maximization (EM). Given this observation, we consider five GMM-based ablations of our method where the log-determinant is introduced (a uniform class prior is used). These ablations are presented in Table 3 and correspond to their soft k-means counterparts in the same order shown. The GMM-based variations of our method and Simple CNAPS result in a notable 4-8% loss in overall accuracy. It's also surprising to observe that the FEOT variation matches the performance of the full GMM-EM model.

**Maximum and Minimum Number of Refinements:** In our experiments, we use a minimum number of 2 refinement steps of class parameters, with the maximum set to 4 on the Meta-Dataset and 10 on the mini/tiered-ImageNet benchmarks. We explore the impact of these hyperparameters on the performance on Transductive CNAPS on the Meta-Dataset in Figure 5. As shown, requiring the same number of refinement steps for every task results in suboptimal performance. This is demonstrated by the fact that the peak performance for each minimum number of steps is achieved with larger number of maximum steps, showcasing the importance of allowing different numbers of refinement steps depending on the task. In addition, we observe that as the number of minimum refinement steps increases, the performance improves up to two steps while declining after. This suggests that, unlike Ren et al. (2018) where only a single refinement step leads to the best performance, our Mahalanobis-based approach can leverage extra steps to further refining the class parameters. We do see a decline in performance with a higher number of steps; this suggests that while our refinement criteria can be effective at performance different number of steps depending on the task, it can potentially lead to over-fitting, justifying the need for an accurately chosen maximum number of steps.

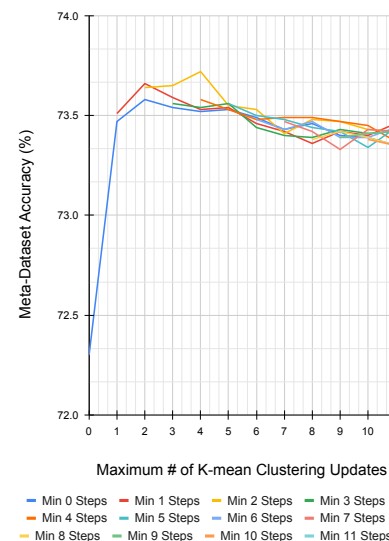

Figure 5: Evaluating Transductive CNAPS on Meta-Dataset with different minimum and maximum number of steps. As shown, performance is improved with 2 minimum refinement steps required, with the best results observed at a maximum of 4 refinement steps. Performance is, however, degraded with more refinement steps required as minimum as that may lead to over-fitting.

## 5 DISCUSSION

In this paper, we have presented a few-shot visual classification method that achieves new state of the art performance via a transductive clustering procedure for refining class parameters derived from a previous neural adaptive Mahalanobis-distance based approach. The resulting architecture, Transductive CNAPS, is more effective at producing useful estimates of class mean and covariance especially in low-shot settings, when used at test time. Even though we demonstrate the efficacy of our approach in the transductive domain where query examples themselves are used as unlabelled data, our soft k-means clustering procedure can naturally extend to use other sources of unlabelled examples in a semi-supervised fashion.

Transductive CNAPS superficially resembles a transductive GMM model stacked on top of a learned feature representation; however, when we try to make this connection exact (by including the log-determinant of the class covariances), we suffer substantial performance hits. Explaining why this happens will be the subject of future work.

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
