# OpenReview forum: "Improving Few-Shot Visual Classification with Unlabelled Examples"
_ICLR.cc/2021/Conference — Reject_

### Official Review · AnonReviewer2 · 2020-10-26
**Please find below for the detailed comments**

**Rating:** 5
**Confidence:** 4

**Review:**

This paper proposes to extend the simple CNAPS few-shot learning method to the transductive setting. Specifically, the proposed method introduces transductive task encoding and soft k-means iterative estimation to improve CNAPS. The proposed method is able to achieve SOTA results on the transductive benchmarks.

+ The proposed method can effectively improve the CNAPS as it achieves SOTA or comparable results on Meta-dataset, mini-ImageNet and tiered-ImageNet.
+ The writing is clear and easy to follow.

- The proposed method lacks enough novelty. The proposed method is actually a transductive extension of CNAPS. The authors introduce two novel components: transductive task encoding and soft k-means clustering. The former is a natural and straight-forward solution while the latter is also similar to previous work [1]. This weakens the paper's overall contribution.
- The experiments could be improved. As stated in the experiment section, some results are not well-prepared before the submission, e.g. Overlapping-excluded ImageNet results.
- According to Fig.4, the authors could further analyze why transductive updates can not remain improvements as the shot increases. Is there a way to design an adaptive scheme of deciding whether to use or not?
[1] Mengye Ren, Eleni Triantafillou, Sachin Ravi, Jake Snell, Kevin Swersky, Joshua B. Tenenbaum,
Hugo Larochelle, and Richard S. Zemel. Meta-learning for semi-supervised few-shot classification

---

> ### Author Response · Authors · 2020-11-18
> **Addressing Specific Questions/Concerns**
>
> > The experiments could be improved. As stated in the experiment section, some results are not well-prepared before the submission, e.g. Overlapping-excluded ImageNet results.
>
> We have updated mini- and tiered-ImageNet experiments considering two different settings as described within our general response. Additional details are provided with our updated paper draft. In short, we have new state of the art performance on all 10-way settings on both benchmarks while matching past state of the art on 5-way settings on tiered-ImageNet.
>
> > According to Fig.4, the authors could further analyze why transductive updates can not remain improvements as the shot increases. Is there a way to design an adaptive scheme of deciding whether to use or not?
>
> That’s an excellent point! Over the past week, we explored two potential approaches, one where class parameter refinement is not done for classes with n or more support examples (with n = 20, 50, 100), and the otherwhere those class parameters are updated during class refinements but then returned to their original in time for the final estimation of class probabilities for query examples. Neither approach has been able to produce empirically better results than Transductive CNAPS. This can in part due to the fact that, as noted in the Appendix-A.1, the frequency of high-shot classes, with over 20 support sets, is substantially lower than that of low-shot classes. Thus, it’s unclear if small gains on those specific classes can lead to major gains across the whole benchmark. We will compile the results and specification for these approaches and discuss them within the supplemental material within the next few days. Thank you again for this question!

---

> ### Author Response · Authors · 2020-11-18
> **General Response: Additional Results, Clarifying Language Confusion, Discussion Technical Contributions**
>
> Thank you very much for your feedback. We really appreciate the concerns and questions brought up. Your comments have helped us improve our submission in a number of ways. In the interest of conciseness, we have separated our response into two sections, first discussing some empirical updates and general concerns brought up by two or more reviewers, and then addressing specific questions/concerns.
>
> General Response:
>
> After reading the reviews, we realized the usage of the word “pre-training” when referring to how the feature extractor is trained has been causing some confusion. At a high level, Transductive CNAPS is trained in two stages. First, the feature extractor is trained as a supervised image classifier using training examples. Then, the feature extractor weights are held fixed, while the adaptation network is trained via episodic training on a large set of few-shot tasks. We have referred to the former as pre-training the ResNet18 in our submission. Recognizing the confusion this may have caused, we have since edited the relevant sections, updating the language and making the process more clear.
>
> In addition, there were some concerns regarding the usefulness and strengths of our results on the mini- and tiered-ImageNet benchmarks. In our initial submission, we considered two settings: one where the feature extractor is trained on CIFAR100 and the other where a test-set overlapping subset of ImageNet was used for training the feature extractor. The former only provided competitive but non-SoTA performance and the latter, could only be compared to other CNAPS based methods (which use the same fixed feature extractor), due to the class/example overlap with the test splits of mini- and tiered- ImageNet. To alleviate these issues, we are now considering two different settings and have updated Table 2 with the new results. First, we consider the case of training the feature extractor on the training set of mini-/tiered-ImageNet. This setting uses no external information and therefore provides a fair comparison with past methods. In this setting, Transductive CNAPS not only provides better results, but is now able to achieve new state of the art performance on 1/5-shot 10-way tasks on both benchmarks while matching state of the art on the 1/5-shot 5 way tasks on tiered-ImageNet. Second, we train the feature extractor on ImageNet excluding classes that overlap with the test sets of mini- and tiered-ImageNet. As shown by the results in Table 2, Transductive CNAPS in this setting is able to achieve state of the art results across the board on mini- and tiered-ImageNet with strong margins, even compared to Simple CNAPS using the same feature extractor.
>
> Furthermore, we would like to address some concerns regarding the technical novelty of our work and its relations to Ren et al. While our work shares the same high-level idea to use unlabelled examples within a k-means clustering framework to update class parameters, it is different in several ways. First, the usage of the two-step task encoder with an example weighted support embedding, provides an effective way to transductively adapt the feature extractor. The use of the LSTM in this process is in part to implicitly model the dependency of query examples on support examples. Second, our method iteratively refines the class covariance estimates; Ren et. al uses (effectively) fixed class covariances in their experiments, with the exception of their “confuser” class. Third, where our approach deviates from that of Ren et al. most significantly, is the use of an iterative k-mean clustering algorithm with a termination criteria and pre-defined minimum and maximum numbers of cluster updates. Unlike our work, Ren et al. only performs a single refinement step. As they describe in the paper, their method can be extended to do more, but the performance declines as the number of refinement steps increases. We instead define a query label assignment criteria that allows different numbers of refinement depending on the task at hand. This, as explored within a newly added experiment section named “Maximum and Minimum Number of Refinements” under section 4.2 in our draft, proves to boost performance. We additionally predefine minimum and maximum numbers of steps, allowing for a range of refinement steps to be possible. Although these details may seem incremental when considered individually, in aggregate they do differ appreciably from previous work, and those differences produce strong empirical results; thus we believe that our work provides greater novelty and technical contribution than some reviewers have suggested.

---

### Official Review · AnonReviewer4 · 2020-10-29
**incremental contribution**

**Rating:** 5
**Confidence:** 3

**Review:**

This paper proposes a transductive few-shot learning method, Transductive CNAPS, by using the unlabeled examples. Experimental results on the Meta-Dataset and mini/tiered-ImageNet benchmarks are reported.

The proposed approach is a simple extension of the simple CNAPS method. It extends the simple CNAPS by using the unlabeled query instances to update the class centroids \mu_k and variance Q_k with the predicted class probabilities. The level of technical contribution and novelty is very incremental and low.

The experiments are not very convincing. First, for both experiments on Meta-Dataset and min/tiered-ImageNet, the feature extractor is pretrained on some ImageNet subset. This is not appropriate as it borrows significant information from the pretraining dataset. The authors also mentioned this issue.
Second, the transductive few-shot learning methods need to be compared with on the Meta-Dataset in Table 1.  Even with the current comparison methods, the performance gains of the proposed approach are not very notable.

Some additional questions:
1.	In Table 1, are all the results obtained using exactly the same sets of training/testing tasks for all comparision methods? What are the source domains for each of the out-of-domain task?
2.	Do all the comparision methods perform the same pre-training?

---

> ### Author Response · Authors · 2020-11-18
> **Answering Specific Questions/Concerns**
>
> > The experiments are not very convincing. First, for both experiments on Meta-Dataset and min/tiered-ImageNet, the feature extractor is pretrained on some ImageNet subset. This is not appropriate as it borrows significant information from the pretraining dataset. The authors also mentioned this issue. Second, the transductive few-shot learning methods need to be compared with on the Meta-Dataset in Table 1. Even with the current comparison methods, the performance gains of the proposed approach are not very notable.
>
> We recognize the confusion that may have been caused by using the word pre-training. As mentioned in the general response, we have updated the language to make this more clear. The feature extractor is trained on the Meta-Dataset defined ImageNet training-set. This only uses the same examples that are also available for episodic training and does not use external examples or overlap with any of the test/validation data. Our experiments on mini/tiered-ImageNet did suffer from the class overlap issue but we have since taken the steps to perform those experiments using feature extractor training on 1. Only the benchmark provided training data and 2. non-test-set overlapping additional examples. For a discussion of these in detail alongside new results on the mini/tiered-ImageNet benchmarks, please refer to the updated experimental section in our updated submission.
>
> > Some additional questions:
> In Table 1, are all the results obtained using exactly the same sets of training/testing tasks for all comparision methods? What are the source domains for each of the out-of-domain task?
> Do all the comparision methods perform the same pre-training?
>
> 1. Not the exact same task. They are sampled from the same data, and the provided confidence intervals, both for our methods and the baselines, account for any potential noise arising from the tasks being sampled within a 95% confidence interval. Sampling test tasks is standard practice and with a high number of sampled test tasks, 600 tasks in our case, the variance is significantly reduced and accounted for within the error intervals.
>
> 2. As mentioned in our general response, the training of the feature extractor is (with the exception of the second ResNet training setting on mini/tiered-imagenet) completely done on the training data of each benchmark. All baselines have had access to the same data and could have done directed supervised training of the feature extractor. However, to our knowledge, only CNAPS based methods, such as CNAPS, Simple CNAPS and our work Transductive CNAPS, only do it in this explicit manner.

---

> ### Author Response · Authors · 2020-11-18
> **General Response: Additional Results, Clarifying Language Confusion, Discussion Technical Contributions**
>
> Thank you very much for your review. We’ve used your feedback to update our paper submission. In the interest of conciseness, we have separated our response into two sections, first discussing some empirical updates and general concerns brought up by two or more reviewers, and then addressing specific questions/concerns.
>
> General Response:
>
> After reading the reviews, we realized the usage of the word “pre-training” when referring to how the feature extractor is trained has been causing some confusion. At a high level, Transductive CNAPS is trained in two stages. First, the feature extractor is trained as a supervised image classifier using training examples. Then, the feature extractor weights are held fixed, while the adaptation network is trained via episodic training on a large set of few-shot tasks. We have referred to the former as pre-training the ResNet18 in our submission. Recognizing the confusion this may have caused, we have since edited the relevant sections, updating the language and making the process more clear.
>
> In addition, there were some concerns regarding the usefulness and strengths of our results on the mini- and tiered-ImageNet benchmarks. In our initial submission, we considered two settings: one where the feature extractor is trained on CIFAR100 and the other where a test-set overlapping subset of ImageNet was used for training the feature extractor. The former only provided competitive but non-SoTA performance and the latter, could only be compared to other CNAPS based methods (which use the same fixed feature extractor), due to the class/example overlap with the test splits of mini- and tiered- ImageNet. To alleviate these issues, we are now considering two different settings and have updated Table 2 with the new results. First, we consider the case of training the feature extractor on the training set of mini-/tiered-ImageNet. This setting uses no external information and therefore provides a fair comparison with past methods. In this setting, Transductive CNAPS not only provides better results, but is now able to achieve new state of the art performance on 1/5-shot 10-way tasks on both benchmarks while matching state of the art on the 1/5-shot 5 way tasks on tiered-ImageNet. Second, we train the feature extractor on ImageNet excluding classes that overlap with the test sets of mini- and tiered-ImageNet. As shown by the results in Table 2, Transductive CNAPS in this setting is able to achieve state of the art results across the board on mini- and tiered-ImageNet with strong margins, even compared to Simple CNAPS using the same feature extractor.
>
> Furthermore, we would like to address some concerns regarding the technical novelty of our work and its relations to Ren et al. While our work shares the same high-level idea to use unlabelled examples within a k-means clustering framework to update class parameters, it is different in several ways. First, the usage of the two-step task encoder with an example weighted support embedding, provides an effective way to transductively adapt the feature extractor. The use of the LSTM in this process is in part to implicitly model the dependency of query examples on support examples. Second, our method iteratively refines the class covariance estimates; Ren et. al uses (effectively) fixed class covariances in their experiments, with the exception of their “confuser” class. Third, where our approach deviates from that of Ren et al. most significantly, is the use of an iterative k-mean clustering algorithm with a termination criteria and pre-defined minimum and maximum numbers of cluster updates. Unlike our work, Ren et al. only performs a single refinement step. As they describe in the paper, their method can be extended to do more, but the performance declines as the number of refinement steps increases. We instead define a query label assignment criteria that allows different numbers of refinement depending on the task at hand. This, as explored within a newly added experiment section named “Maximum and Minimum Number of Refinements” under section 4.2 in our draft, proves to boost performance. We additionally predefine minimum and maximum numbers of steps, allowing for a range of refinement steps to be possible. Although these details may seem incremental when considered individually, in aggregate they do differ appreciably from previous work, and those differences produce strong empirical results; thus we believe that our work provides greater novelty and technical contribution than some reviewers have suggested.

---

### Official Review · AnonReviewer3 · 2020-10-29
**Interesting work but lacks novelty and performance improvements in some cases are limited**

**Rating:** 6
**Confidence:** 3

**Review:**

In order to improve few shot visual classification, the authors propose a transductive meta-learning method using unlabelled examples. The authors have introduced a two-step transductive encoder as well as soft k-means clustering procedure on the existing simple CNAPS architecture.

Pros:
The paper is well written.
The authors have improved upon the existing simple CNAPS model by making two changes:
i) Simple CNAPS extracted feature vector by passing support example through a CNN to get support task representation $g_\theta(S)$ which is given as an input to FiLM layers to learn scale and shift parameters. This produces an adapted feature extractor to map both support/query onto adapted feature space. While in transductive SNAPS the support task representation is used to compute support-set embedding ($e_s$) and query-set embedding ($e_q$). Both $e_s$ and $e_q$ are processed by a two-step LSTM to generate the final transductive task-embedding used for adaptation.
ii) Introduce an iterative algorithm to assign soft-label assignment of query image which in turn performs iterative estimation of class means and covariances.

Cons:
i) The performance on Meta-Dataset in Table 1 seems to show that the average rank is better only in the case for the Out-of-Domain test case.
ii) Table 2 shows when Transductive CNAPS compared with other transductive approaches the performance is not that great. However, when it is compared with other SNAPS architecture the performance is good. The missing experiment of excluding the overlapping classes would provide better insight into it.
iii) As already mentioned by the authors, this work resembles Gaussian Mixture Models.

Overall the model's novelty over the existing Simple SNAPS architecture is limited. Even the quantitate scores reported are not superior enough; for example in the case of Meta-Dataset the model suffers in In-Domain accuracy and does not achieve SOTA in case of mini/tiered-ImageNet.

Final review
The authors have addressed some of my concerns so I am updating the rating. I still feel that the mode's novelty is limited and thus my highest rating will be 6.

---

> ### Author Response · Authors · 2020-11-18
> **Addressing Specific Questions/Concerns**
>
> > i) The performance on Meta-Dataset in Table 1 seems to show that the average rank is better only in the case for the Out-of-Domain test case. ii) Table 2 shows when Transductive CNAPS compared with other transductive approaches the performance is not that great. However, when it is compared with other SNAPS architecture the performance is good. The missing experiment of excluding the overlapping classes would provide better insight into it. iii) As already mentioned by the authors, this work resembles Gaussian Mixture Models.
>
> i) The average rank of Meta-Dataset is better on both the out-of-domain tasks and overall. In addition, the only method that outperforms Transductive CNAPS on in-domain rank is URT, which builds on the SUR baseline. Both of these methods use domain-specific trained ResNet feature extractor for each of the 8 in-domain datasets. This allows them to resort to any of these ResNet heads when needed, and therefore, perform very well on in-domain tasks. Our method on the other hand, performs such training only on the ImageNet subset of Meta-Dataset. We believe that while their approach may be effective at producing strong in-domain performance, it is overly specialized to the Meta-Dataset setting, and as demonstrated by their out-of-domain performance, it suffers from generalizing to out-of-domain tasks, which we believe is a fundamental goal in few-shot learning.
>
> ii) As noted in our general response, we have updated those experiments, reporting new state of the art performance on 4/8 settings on these benchmarks, 2/8 matching past state of the art, when deployed with a more feature-rich ResNet, Transductive CNAPS achieves state of the art performance across the board. Please refer to the mini-/tiered-ImageNet section of our updated draft for the new results and their detailed discussion.
>
> iii) It’s unclear to us why the relation to gaussian mixture modelling is considered a downside. We believe that not only this relation provides some theoretical grounding for our work, but also our results showcase an interesting point where within our setup, the inclusion of the log-determinant normalizer results in worse performance contrary to the expectation. We hope that future exploration of this observation can lead to interesting finding on gaussian mixture model in high dimensions.

---

> ### Author Response · Authors · 2020-11-18
> **General Response: Additional Results, Clarifying Language Confusion, Discussion Technical Contributions**
>
> We really appreciate your feedback and have taken steps to address your concerns both here in our response and within our updated submission. In the interest of conciseness, we have separated our response into two sections, first discussing some empirical updates and general concerns brought up by two or more reviewers, and then addressing specific questions/concerns.
>
> General Response:
>
> After reading the reviews, we realized the usage of the word “pre-training” when referring to how the feature extractor is trained has been causing some confusion. At a high level, Transductive CNAPS is trained in two stages. First, the feature extractor is trained as a supervised image classifier using training examples. Then, the feature extractor weights are held fixed, while the adaptation network is trained via episodic training on a large set of few-shot tasks. We have referred to the former as pre-training the ResNet18 in our submission. Recognizing the confusion this may have caused, we have since edited the relevant sections, updating the language and making the process more clear.
>
> In addition, there were some concerns regarding the usefulness and strengths of our results on the mini- and tiered-ImageNet benchmarks. In our initial submission, we considered two settings: one where the feature extractor is trained on CIFAR100 and the other where a test-set overlapping subset of ImageNet was used for training the feature extractor. The former only provided competitive but non-SoTA performance and the latter, could only be compared to other CNAPS based methods (which use the same fixed feature extractor), due to the class/example overlap with the test splits of mini- and tiered- ImageNet. To alleviate these issues, we are now considering two different settings and have updated Table 2 with the new results. First, we consider the case of training the feature extractor on the training set of mini-/tiered-ImageNet. This setting uses no external information and therefore provides a fair comparison with past methods. In this setting, Transductive CNAPS not only provides better results, but is now able to achieve new state of the art performance on 1/5-shot 10-way tasks on both benchmarks while matching state of the art on the 1/5-shot 5 way tasks on tiered-ImageNet. Second, we train the feature extractor on ImageNet excluding classes that overlap with the test sets of mini- and tiered-ImageNet. As shown by the results in Table 2, Transductive CNAPS in this setting is able to achieve state of the art results across the board on mini- and tiered-ImageNet with strong margins, even compared to Simple CNAPS using the same feature extractor.
>
> Furthermore, we would like to address some concerns regarding the technical novelty of our work and its relations to Ren et al. While our work shares the same high-level idea to use unlabelled examples within a k-means clustering framework to update class parameters, it is different in several ways. First, the usage of the two-step task encoder with an example weighted support embedding, provides an effective way to transductively adapt the feature extractor. The use of the LSTM in this process is in part to implicitly model the dependency of query examples on support examples. Second, our method iteratively refines the class covariance estimates; Ren et. al uses (effectively) fixed class covariances in their experiments, with the exception of their “confuser” class. Third, where our approach deviates from that of Ren et al. most significantly, is the use of an iterative k-mean clustering algorithm with a termination criteria and pre-defined minimum and maximum numbers of cluster updates. Unlike our work, Ren et al. only performs a single refinement step. As they describe in the paper, their method can be extended to do more, but the performance declines as the number of refinement steps increases. We instead define a query label assignment criteria that allows different numbers of refinement depending on the task at hand. This, as explored within a newly added experiment section named “Maximum and Minimum Number of Refinements” under section 4.2 in our draft, proves to boost performance. We additionally predefine minimum and maximum numbers of steps, allowing for a range of refinement steps to be possible. Although these details may seem incremental when considered individually, in aggregate they do differ appreciably from previous work, and those differences produce strong empirical results; thus we believe that our work provides greater novelty and technical contribution than some reviewers have suggested.

---

### Official Review · AnonReviewer1 · 2020-10-30
**This paper proposes a transductive few-shot classification method on the basis of the simple Conditional Neural Adaptive Processes (CNAPS) introduced by Bateni et al. The proposed method, called transductive CNAPS, extends the simple CNAPS by exploiting the query set both in the feature adaptation stage and classification stage. Extensive experiments are conducted on Meta-Dataset, mini-ImageNet and tiered-ImageNet, and very competitive results are reported.**

**Rating:** 6
**Confidence:** 3

**Review:**

Pros:

1) The paper is well-motivated, clearly written and neatly organized. I like the writing style of this paper: explaining very clearly what the improvements are and how they perform with respect to the counterpart methods.

2) The ablation study validates the effectiveness of the improved adaptation of the feature extractor using transductive task-encoding, and that of the soft k-means iterative estimation of means and covariances for classification.

3) Experiments are solid and extensive. The results on Meta-Dataset, mini/tiered-ImageNet are competitive, as opposed to the state-of-the-art methods.

Cons:

1) The originality of the proposed method is incremental. Compared to simple CNAPS (Betni et al, CVPR 2020), the task encoder is extended to incorporate both a support-set embedding and a query-set embedding through Long-Short Term memory (LSTM) network, and the classifier is extended to include the unlabeled examples in the query set as well. As far as I know, encoding both support examples and query ones using LSTM has been used in Matching Network (Vinyals et al., 2016) while the idea of using query examples to refine class porotypes via soft k-means clustering were proposed by Ren et al. ,2018.

2)  Regarding the soft-kmean iterative estimation, why can it be only used at test time while the performance decreases significantly if used during training as well? How many iterations are performed? Will various number of iterations matter? In particular, I am not satisfied with the limited explanation on this method provided in Section 5.2 (last paragraph on page 7), and I wish the authors give more analysis and insight on this problem.

3)  Some minor points:
In Table 2, what does "BN” mean?
How to set the value of $\beta$ in Equation (6)?
Some typos: In Section 3.2, "metalearning framework "  --> "meta learning framework "; in Section 4.2, "ou-of-domain" --> "out-of-domain".

---

> ### Author Response · Authors · 2020-11-18
> **General Response: Additional Results, Clarifying Language Confusion, Discussion Technical Contributions**
>
> Let us first thank you for taking the time to provide a detailed review of our work. We really appreciate your feedback as it has helped us improve the paper over the past week in the run-up to this rebuttal. In the interest of conciseness, we have separated our response into two sections, first discussing some empirical updates and general concerns brought up by two or more reviewers, and then addressing specific questions/concerns.
>
> General Response:
>
> After reading the reviews, we realized the usage of the word “pre-training” when referring to how the feature extractor is trained has been causing some confusion. At a high level, Transductive CNAPS is trained in two stages. First, the feature extractor is trained as a supervised image classifier using training examples. Then, the feature extractor weights are held fixed, while the adaptation network is trained via episodic training on a large set of few-shot tasks. We have referred to the former as pre-training the ResNet18 in our submission. Recognizing the confusion this may have caused, we have since edited the relevant sections, updating the language and making the process more clear.
>
> In addition, there were some concerns regarding the usefulness and strengths of our results on the mini- and tiered-ImageNet benchmarks. In our initial submission, we considered two settings: one where the feature extractor is trained on CIFAR100 and the other where a test-set overlapping subset of ImageNet was used for training the feature extractor. The former only provided competitive but non-SoTA performance and the latter, could only be compared to other CNAPS based methods (which use the same fixed feature extractor), due to the class/example overlap with the test splits of mini- and tiered- ImageNet. To alleviate these issues, we are now considering two different settings and have updated Table 2 with the new results. First, we consider the case of training the feature extractor on the training set of mini-/tiered-ImageNet. This setting uses no external information and therefore provides a fair comparison with past methods. In this setting, Transductive CNAPS not only provides better results, but is now able to achieve new state of the art performance on 1/5-shot 10-way tasks on both benchmarks while matching state of the art on the 1/5-shot 5 way tasks on tiered-ImageNet. Second, we train the feature extractor on ImageNet excluding classes that overlap with the test sets of mini- and tiered-ImageNet. As shown by the results in Table 2, Transductive CNAPS in this setting is able to achieve state of the art results across the board on mini- and tiered-ImageNet with strong margins, even compared to Simple CNAPS using the same feature extractor.
>
> Furthermore, we would like to address some concerns regarding the technical novelty of our work and its relations to Ren et al. While our work shares the same high-level idea to use unlabelled examples within a k-means clustering framework to update class parameters, it is different in several ways. First, the usage of the two-step task encoder with an example weighted support embedding, provides an effective way to transductively adapt the feature extractor. The use of the LSTM in this process is in part to implicitly model the dependency of query examples on support examples. Second, our method iteratively refines the class covariance estimates; Ren et. al uses (effectively) fixed class covariances in their experiments, with the exception of their “confuser” class. Third, where our approach deviates from that of Ren et al. most significantly, is the use of an iterative k-mean clustering algorithm with a termination criteria and pre-defined minimum and maximum numbers of cluster updates. Unlike our work, Ren et al. only performs a single refinement step. As they describe in the paper, their method can be extended to do more, but the performance declines as the number of refinement steps increases. We instead define a query label assignment criteria that allows different numbers of refinement depending on the task at hand. This, as explored within a newly added experiment section named “Maximum and Minimum Number of Refinements” under section 4.2 in our draft, proves to boost performance. We additionally predefine minimum and maximum numbers of steps, allowing for a range of refinement steps to be possible. Although these details may seem incremental when considered individually, in aggregate they do differ appreciably from previous work, and those differences produce strong empirical results; thus we believe that our work provides greater novelty and technical contribution than some reviewers have suggested.

---

> ### Author Response · Authors · 2020-11-18
> **Answering Specific Questions**
>
> > Regarding the soft-kmean iterative estimation, why can it be only used at test time while the performance decreases significantly if used during training as well? How many iterations are performed? Will various number of iterations matter? In particular, I am not satisfied with the limited explanation on this method provided in Section 5.2 [4.2] (last paragraph on page 7), and I wish the authors give more analysis and insight on this problem.
>
> The refinement steps rely on a feature space where soft-labels can be useful to refining the class parameters. This is especially difficult to obtain in the early stages of training and can therefore result in non-related query examples moving class parameters around which can in turn produce suboptimal gradients when training the adaptation network. This is demonstrated by the discrepancy between the performance when the refinement procedure is applied during training and when it’s not. We use a refinement criteria of query examples not changing predicted (max likelihood) label as the soft-labels are updated. This allows for different number of iterations performed depending on the task at hand. We also apply a minimum and maximum number of iterations required. These hyperparameters which effectively currently the number of iterations are also explored through a grid range within a newly added section under 4.2,  “Maximum and Minimum Number of Refinements”, providing detailed insights into Transductive CNAPS’ behaviour with respect to number of refinement steps.
>
> > Some minor points: In Table 2, what does "BN” mean? How to set the value of β in Equation (6)? Some typos: In Section 3.2, "metalearning framework " --> "meta learning framework "; in Section 4.2, "ou-of-domain" --> "out-of-domain".
>
> BN stands for batch-normalization; it is noted because, while the baselines do not explicitly use transductive learning using the query examples, batch norm acts as a sort of implicit transduction, in the sense that the resulting classifier is a function of both the support and query sets. Following Simple CNAPS, we also set β=1. Thank you for pointing out the typos. They have been fixed.
>
> We thank you again for your feedback and have updated the paper submission accordingly!

---

### Decision · Program_Chairs · 2021-01-07
**Final Decision**

**Decision:**

Reject

**Comment:**

The submission proposes a transductive few-shot classification method on the basis of the simple Conditional Neural Adaptive Processes (CNAPS) introduced by Bateni et al. The paper received two borderline accept and two borderline reject reviews, indicating that the paper may not be yet ready for a publication. The meta reviewer recommends rejection based on the observations below.

All reviewers indicated that the paper is well-motivated, clearly written and neatly organized. However, all four reviewers agree that the novelty of the paper compared to the CNAPS paper is limited. The main novelty of the method being transductive CNAPS extends the task encoder of CNAPS to incorporate both a support-set embedding and a query-set embedding through Long-Short Term memory (LSTM) network. Similarly, the classifier in CNAPS has been modified to operate in the transduction setting, i.e. it is extended to include the unlabeled examples in the query set.  The reviewers indicate that extension of the task encoder via LSTM may not be enough technical novelty for such a competitive venue. Additionally, in terms of experimental evaluation, although R1 found the experimental evaluation adequate, R3 indicated some concerns about the unexpected behaviour of the method and R4&R2 found the benchmark evaluations limited.